# *Lactobacillus gasseri* LG-G12 Restores Gut Microbiota and Intestinal Health in Obesity Mice on Ceftriaxone Therapy

**DOI:** 10.3390/foods12051092

**Published:** 2023-03-03

**Authors:** Mariana de Moura e Dias, Vinícius da Silva Duarte, Lúcio Flávio Macedo Mota, Gabriela de Cássia Ávila Alpino, Sandra Aparecida dos Reis Louzano, Lisiane Lopes da Conceição, Hilário Cuquetto Mantovanie, Solange Silveira Pereira, Leandro Licursi Oliveira, Tiago Antônio de Oliveira Mendes, Davide Porcellato, Maria do Carmo Gouveia Peluzio

**Affiliations:** 1Department of Nutrition and Health, Universidade Federal de Vicosa, Avenida P. H. Rolfs, Campus Universitário S/N, Viçosa 36570-900, Minas Gerais, Brazil; 2Faculty of Chemistry, Biotechnology, and Food Science, The Norwegian University of Life Sciences, P.O. Box 5003, N-1432 Ås, Norway; 3Department of Agronomy Food Natural Resources Animals and Environment, University of Padova, Viale dell’Università, 16, 35020 Padua, Italy; 4Department of Microbiology, Universidade Federal de Vicosa, Avenida P. H. Rolfs, Campus Universitário S/N, Viçosa 36570-900, Minas Gerais, Brazil; 5Department of General Biology, Universidade Federal de Vicosa, Avenida P. H. Rolfs, Campus Universitário S/N, Viçosa 36570-900, Minas Gerais, Brazil; 6Department of Biochemistry and Molecular Biology, Universidade Federal de Vicosa, Avenida P. H. Rolfs, Campus Universitário S/N, Viçosa 36570-900, Minas Gerais, Brazil

**Keywords:** *Lactobacillus gasseri* LG-G12, intestinal health, ceftriaxone, gut microbiota, obesity

## Abstract

Gut microbiota imbalance is associated with the occurrence of metabolic diseases such as obesity. Thus, its modulation is a promising strategy to restore gut microbiota and improve intestinal health in the obese. This paper examines the role of probiotics, antimicrobials, and diet in modulating gut microbiota and improving intestinal health. Accordingly, obesity was induced in C57BL/6J mice, after which they were redistributed and fed with an obesogenic diet (intervention A) or standard AIN-93 diet (intervention B). Concomitantly, all the groups underwent a treatment phase with *Lactobacillus gasseri* LG-G12, ceftriaxone, or ceftriaxone followed by *L. gasseri* LG-G12. At the end of the experimental period, the following analysis was conducted: metataxonomic analysis, functional profiling of gut microbiota, intestinal permeability, and caecal concentration of short-chain fatty acids. High-fat diet impaired bacterial diversity/richness, which was counteracted in association with *L. gasseri* LG-G12 and the AIN-93 diet. Additionally, SCFA-producing bacteria were negatively correlated with high intestinal permeability parameters, which was further confirmed via functional profile prediction of the gut microbiota. A novel perspective on anti-obesity probiotics is presented by these findings based on the improvement of intestinal health irrespective of undergoing antimicrobial therapy or not.

## 1. Introduction

The gut is home to approximately 70% of the microbiota detected in humans, including bacteria, fungi, viruses, and protozoa [1,2]. Modulation of gut microbiota composition and metabolic functions have been proposed as key factors that control obesity development [2,3].

External modulators of gut microbiota include probiotics, antimicrobials, and diet, which acts with speed and high precision in order to impact obesity [4,5]. Probiotics are live microorganisms which provide health benefits to host when consumed in sufficient amounts. They modulate the composition and metabolic functions of the gut microbiota, and contribute to immunological functions through the regulation of cytokines, promotion of oral tolerance to food antigens, and improvement of intestinal barrier functions [5]. In this context, several probiotics, used alone or as synbiotic mixtures, have shown antiobesity effects. For example, *Lactobacillus gasseri* has beneficial effects on weight loss and body fat reduction in overweight humans and animals [6,7,8].

Alternatively, antimicrobials can contribute to the loss of microbial diversity in the gut over time, impairing metabolic function and leading to impaired metabolism. Therefore, they potentially reduce the colonization resistance against invading pathogens, resulting in dysbiosis [9]. Antimicrobial’s ability to alter the microbiota of the gut varies based on diet, lifestyle, and drug spectrum of action as well as its absorption capacity, which indicates that broader spectrum antimicrobials such as ceftriaxone result in intestinal dysbiosis more frequently [10,11].

Similarly, a high-fat diet (HFD) is capable of promoting intestinal dysbiosis, thus contributing to intestinal barrier dysfunction, immune intolerance to food antigens, activation of pro-inflammatory routes, and circadian cycle disruption that leads to weight gain, abnormal glucose fluxes, and inflammatory response [12]. In contrast, balanced and diversified diets, such as the Mediterranean standard diet, rich in fruits, vegetables, whole grains, and seafood, promotes a diverse gut microbiota, thereby stimulating intestinal barrier function and immunity [13].

Moreover, the production of short-chain fatty acids (SCFAs), metabolites of gut microbiota, and intestinal permeability are also indicative of intestinal and systemic health [14,15]. SCFAs have numerous beneficial effects on the host from increasing mucus and tight junction expressions in the intestinal epithelium to metabolic and appetite modulation [16]. Intestinal permeability is influenced by gut microbiota imbalance and is one of the main factors for low-grade inflammation, making the microbiome a central player as regards inflammatory diseases such as obesity [12].

Given the above, there is a need for insights into mechanisms used to regulate dysbiosis in obese mice. Therefore, this study evaluated how a potential probiotic *Lactobacillus gasseri* LG-G12 (LG-G12), antimicrobial ceftriaxone, and diet [7,8] act to modulate the intestinal microbiota and subsequently impact intestinal health parameters.

## 2. Materials and Methods

### 2.1. Animals

Animal Ethics Committee of Universidade Federal de Viçosa approved the experiment according to the protocols numbers 09/2017 and 33/2018, and the principles established by the National Animal Experimentation Control Council [17].

An experiment was conducted using 72 male C57BL/6J mice [7,8] from the Central Vivarium of the Center for Biological and Health Sciences of Universidade Federal de Viçosa (UFV). The mice were kept in collective cages (two animals per cage) and were submitted to a 12 h light/dark cycle and an average temperature of 22 ± 2 °C. A pair-feeding scheme was used to administer fructose solution and a diet to the animals throughout the experiment. All animal experiments were conducted at UFV’s Experimental Nutrition Laboratory.

### 2.2. Experimental Design and Diet

At 5 weeks, the animals underwent an obesity-induced protocol that lasted 3 months (induction phase). During this initial period, the mice were fed with an HFD where 60% of total calories were derived from lipids (nutritional composition based on diet D12492 of the Research Diets, Inc., New Brunswick, NJ, USA) [18], and a 10% fructose solution (Synth^®^, Diadema, Brazil) instead of drinking water [19]. The negative control group (G1, n = 8) received an AIN-93 diet, with 10% of total calories derived from lipids [20] and water from the induction phase until the end of the experiment.

After this period, the treatment phase began and the mice that were fed with an HFD were randomly divided into two intervention groups (A and B) with a subset of four experimental groups each (Figure 1): high-fat diet (HFD) (G2, n = 7), LG-G12 HFD (G3, n = 7), cefriatoxane HFD (G4, n = 7), cefriatoxane + LG-G12 HFD (G5, n = 7), standard fat diet (SFD) (G6, n = 8), LG-G12 SFD (G7, n = 7), cefriatoxane SFD (G8, n = 7), and cefriatoxane + LG-G12 SFD (G9, n = 8). Intervention A groups continually received an obesogenic diet during the treatment phase. The obesogenic diet was confirmed by Dias et al. [7]. In contrast, intervention B groups initiated an AIN-93 standard diet and water instead of fructose solution during the treatment phase. A gavage treatment was administered every evening at the same time.

The antimicrobial group was treated with 500 mg of ceftriaxone per kg of body mass (Triaxton, Blau Farmacêutica S/A^®^, Cotia, Brazil) [21]. The potential probiotic group received 10^9^ colony-forming units (CFU) of LG-G12 (Lemma Supply Solutions^®^, São Paulo, Brazil). *L. gasseri* LG-G12 was lyophilized and diluted into 200ul of water. Both treatment groups received 500 mg of ceftriaxone per kg of body weight in the first two weeks of the treatment phase and, in the following two weeks, 10^9^ CFU of *L. gasseri* LG-G12.

At the conclusion of the treatment phase, a total exsanguination was carried out after the animals had been anesthetized with 3% isoflurane (Cristália^®^, Belo Horizonte, Brazil) and euthanized. This form of euthanasia is recommended for rodents by CONSEA [17]. For future analyses, tissue samples were collected and stored. More information about the experimental design is available at Dias et al. [7].

### 2.3. Intestinal Permeability

As previously described by Dias et al. [7], after the treatment phase, a lactulose (Daiichi Sankyo^®^, Barueri, Brazil) and mannitol solution (Synth^®^, Diadema, Brazil) were supplied to the animals. Subsequently, the 24 h urine of the animals was collected. These sugars were measured by high-performance liquid chromatography (detector model RID 10A, Shimadzu^®^, Tokyo, Japan).

### 2.4. Quantification of Short-Chain Fatty Acids

By Siegfried et al. [22] method, the acetic, propionic, and butyric acids, of the cecal content, were determined, as reported by Dias et al. [7], and analyzed by high-performance liquid chromatography (Ultimate 3000, Dionex, Thermo Fisher Scientific^®^, Waltham, MA, USA).

### 2.5. Composition and Functional Prediction of the Intestinal Microbiota

A pool of stool samples was made as described by Dias et al. [7]. This methodology was adopted because the animals are isogenic and live in a controlled environment, and can therefore be considered biological replicates.

Metagenomic DNA was extracted from 200 mg of feces using the method adapted from Zhang et al. [23]. Afterward, the quantity of the extracted DNA was evaluated utilizing Qubit (Invitrogen, Thermo Fisher, USA), whereas its integrity and quality were verified through electrophoresis in 1.8% agarose gel. The V3 and V4 regions of 16S rRNA genes were PCR amplified utilizing specific primers (Bakt 341F and Bakt 805R) and sequenced using an Illumina MiSeq desktop sequencer (Illumina, San Diego, CA, USA) at the Macrogen Company (Macrogen Inc^®^, Seoul, South Korea).

Microbiota data were processed and analyzed with QIIME2 (version 2020.2) [24]. In brief, raw sequence data obtained across the C57BL/6J mice stool samples from group G1 to G9 were imported via the Casava1.8 paired-end pipeline followed by denoising with DADA2 [25] (via q2-dada2). Subsequently, an amplicon sequence variants (ASV) table was constructed to generate a phylogenetic tree by using the align-to-tree-might-fast tree pipeline from the q2-phylogeny plugin [26,27]. When appropriate, samples were rarefied to a sampling depth of 120,326 sequences. Taxonomy was assigned to the 16S data using a Naïve Bayes pre-trained Greengenes 13_8 99% OTUs classifier [28].

For the functional prediction of the gut microbiota, ASVs (read sequences and read counts) were used as inputs for the PICRUSt2 (Phylogenetic Investigation of Communities by Reconstruction of Unobserved States) pipeline [29]. In brief, ASVs were inserted and aligned into a reference tree composed of 20,000 full 16S rRNA genes from bacterial and archaeal genomes using, respectively, the HMMER (http://www.hmmer.org, accessed on 1 June 2021) and EPA-ng/GAPPA tools [30,31]. Subsequently, the castor R package [32] was used to predict the missing gene families (Enzyme Commission numbers) for each ASV, as well as their respective copy number of 16S rRNA gene sequences, by using the output tree generated in the previous step. Finally, MinPath [33] was adopted to infer MetaCyc pathways based on EC number abundances.

Under accession numbers PRJNA705760 and PRJNA745938, the raw fastq data have been submitted to the Sequence Read Archive (SRA) at NCBI.

### 2.6. Statistical Analysis

The principal component analysis (PCA) was performed using the relative abundance of the most abundant genera (greater than 0.1% in at least one sample). OTU abundance was scaled and then the PCA analysis was performed using the prcomp function of the R program (R Core Team 3.6.2, 2019). Normalization was performed to assure that the PCA results are mathematically independent of the overlap measure. The factorial analysis for OTU information aimed at obtaining the variables which contribute to the highest differentiation across the treatment groups using the FactoMineR R package [34].

Shapiro-Wilk test was used to test the normality of the variables (i.e., Shannon, Chao1, acetate, propionate, butyrate, total SCFA, and lactulose/mannitol ratio). One-way analysis of variance (ANOVA) followed by Tukey’s post hoc test was adopted for parametric data, and Kruskal Wallis complemented by Dunn’s multiple-comparison test was used for non-parametric data. The results were expressed as mean ± standard error of the mean (SEM). Differentially abundant taxa after the treatment phase that most likely explain the differences among the groups (i.e., fecal biomarkers) were assessed using the linear discriminant analysis (LDA) effect size (LEfSe) [35] tool and setting an alpha value of 0.05 and Log LDA threshold of 2.0.

Beta diversity was assessed using the Unweighted and Weighted UniFrac metrics to evaluate bacterial community dissimilarities between the groups. Permutational multivariate analysis of variance (PERMANOVA) with 999 permutations was used to test whether distances between samples within a certain group are more similar to each other or not. Correlations between continuous variables were determined by Pearson’s (parametric data) or Spearman’s (non-parametric data) correlation with the Paleontological statistics software package for education and data analysis (PAST, v4.06b) [36].

Lastly, the Statistical Analysis of Metagenomic Profile (STAMP) software [37] was used to explore and compare the metabolic potential of the predicted microbial communities across the groups. Functional profiling was built based on the MetaCyc Metabolic Pathway Database [38]. Welch’s t-test (two-sided) was adopted as a statistical hypothesis test. For both analyses, a *p*-value less than 0.05 was considered a significant difference.

## 3. Results and Discussion

### 3.1. Multidimensional Scaling Analysis (MDS)

MDS analysis based on intestinal microbiota abundance explains around 38.9% of the variation between the groups (G1 to G9) (Figure 2). In intervention “B” (groups G1, G6, G7, G8, and G9) there is microbial homogeneity between G7 and G9. Then it is believed that LG-G12 acted as a positive modulator of the intestinal microbiota (Figure 2A). Moreover, distinct bacterial genera contributed differently to the total intestinal microbial load across the different experimental groups, which suggests different biological and/or metabolic capabilities (Figure 2B).

### 3.2. Alpha and Beta Diversity

The absence of differences in alpha diversity between interventions “A” and “B” was observed (Table 1, Appendix A). However, in the groups that consumed an antimicrobial and high-fat diet (G4 and G5), a tendency towards a reduction in the Shannon and Chao1 indices could be observed, when compared to those who received ceftriaxone and a standard diet (G8 and G9). This result shows a synergy between the antibiotic and high-fat diet which can impair bacterial diversity/richness. As previously described [39,40], the use of antibiotics can harm the intestinal epithelium, as well as favor the growth of specific microbial taxa. When associated with high-calorie diets, both factors can contribute negatively to the development of the gut microbiota [40,41], which justifies our findings on cefriatoxane with the high-fat diet group (G4). Finally, switching to a standard diet (intervention “B”) was capable of increasing intestinal diversity in all groups, which justifies the lack of statistical difference.

Regarding the beta-diversity analysis, there was no difference in terms of gut microbiota dispersion based on UniFrac distance metrics (weighted UniFrac: F-value = 0.77; *p* = 0.49; unweighted UniFrac: F-value = 1.66; *p*-value = 0.15). According to PERMANOVA results, a statistically significant difference was observed regarding community dissimilarity considering both UniFrac indices (weighted UniFrac: F-value = 2.68; *p*-value = 0.003; unweighted UniFrac: F-value = 1.75; *p*-value = 0.002). Moreover, pairwise comparisons using Qiime beta-group-significance command revealed that the gut composition of the group that received ceftriaxone and a high-fat diet (G4) showed the greatest distance to other groups, especially to G1 (Appendix A). This result indicates that significant differences in gut microbial composition were due to the intervention and not to dispersion effects. Taken together, alpha- and beta-diversity indices evidenced that a synergy between antibiotics and a high-fat diet can impair bacterial community structure and diversity both qualitatively and quantitatively.

### 3.3. Linear Discriminant Analysis Effect Size (LEfSe)

#### 3.3.1. Phylum Level

Overall, 127 significantly discriminative features (LDA > 2, *p* < 0.05) were identified in the LEfSE analysis. The phyla *Proteobacteria* (LDA > 5) and *Spirochaetes* (LDA > 2.0) were enriched in group G3, whereas the phylum *Tenericutes* (LDA > 2.0) can be considered a biomarker for group G4 (Figure 3). The consumption of a processed diet, commonly rich in emulsifiers and artificial sweeteners, can explain the expansion of *Proteobacteria*, which can increase intestinal permeability and reduce local mucus production [42,43]. Regarding the phylum *Spirochaetes*, although its identification in fecal samples has not been associated with obesity, Jabbar et al. [44] reported the association between *Brachyspira* and irritable bowel syndrome. Our results indicate a limited impact of the LG-G12 in controlling the two aforementioned taxa. Since a single strain was used in this study, its inclusion in a multiple strain formulation, as suggested by the World Gastroenterology Organisation [5], might represent a more effective strategy in limiting the growth of such undesired phyla, which must be evaluated in further studies.

Nevertheless, *Firmicutes* and *Bacteroidetes* have not been identified as biomarkers in any of the interventions, group G4 showed a much higher F/B ratio (Table 2), which is common in the obese population [42]. This means that the combined use of ceftriaxone and a high-fat diet disrupts the intestinal bacterial balance, favoring the growth of a few phyla at the expense of others, which is common in intestinal dysbiosis [45]. Interestingly, following LG-G12 administration, the F/B ratio decreased in the cefriatoxane + LG-G12 A group (G5). This outcome suggests a mechanism of counteraction of LG-G12 to the damage caused by ceftriaxone administration favoring the restoration of intestinal homeostasis [45].

The Gram-positive and Gram-negative genera ratio was also evaluated in each group (Table 2). As expected, the G4 group presented the least proportion of Gram-negative among all groups (23.50%) and the greatest G+/G− ratio (3.24), indicating the selectivity of the antimicrobial ceftriaxone against this specific group of bacteria. Ceftriaxone is a third-generation cephalosporin that targets most Gram-negative bacteria inducing changes in gut microbiota [46]. Our results also reveal that the intervention with LG-G12 alleviated the effects of the synergism between the antimicrobial and the diet offered, restoring intestinal Gram-negative taxa at levels similar to the control groups. Crovesy et al. [6], in a systematic review of randomized controlled clinical trials, suggested that the modulatory effect of *Lactobacillus* in weight loss is strain-dependent and can require its association with calorie restriction, phenolic compounds, or other bacterial strains.

#### 3.3.2. Genus Level

Amongst the biomarkers identified by LEfSE analysis, the genera *Oscillospira* (LDA > 4), *Sporosarcina* (LDA > 4), *Allobaculum* (LDA > 3), *Jeotgalicoccus* (LDA > 3), *Bifidobacterium* (LDA > 3), and *Yaniella* (LDA > 3) were assigned as biomarkers for group G1 (Figure 3). The enrichment of the genera *Bifidobacterium* is in agreement with the literature, where an inverse relationship was shown between this genus and obesity. *Bifidobacteria* deconjugate bile acids, decreasing fat absorption [47]. The higher abundance of *Oscillospira* in the gut of lean subjects has been addressed in several studies and is positively associated with lower body mass index (BMI) in both children and adults [48]. It is also well reported that a high-fat diet can significantly reduce the intestinal abundance of *Allobaculum*, a relevant SCFA-producing bacteria, which may display an anti-obesogenic role by reducing intestinal inflammation and improving insulin resistance [49,50,51]. The increase in the relative abundance of the genera *Jeotgalicoccus* and *Sporosarcina*, although less described in the literature, are associated with beneficial outcomes in animal models fed high-fat diets [52,53]. To the best of our knowledge, there is no available information regarding the role of *Yaniella*, a high salt-tolerant microorganism, in healthy or obese subjects. Overall, our results show that low-calorie diets are beneficial to the maintenance of taxa that are negatively associated with obesity.

The genera *Lactobacillus* (LDA > 4), *Dehalobacterium* (LDA > 2), and *Erysipelotrichaceae cc_115* (LDA > 3) were identified as biomarkers in the obese control group (G2). Although most lactobacilli strains can have beneficial and auxiliary effects on weight loss in overweight adults [6], some species, such as *Limosilactobacillus reuteri* have been associated with weight gain in humans and animals [54,55]. Regarding the genera *Dehalobacterium* and *Erysipelotrichaceae cc-115,* little information is available regarding their abundance and role within the intestinal community of obese or overweight subjects. It is reported that the genus *Dehalobacterium* comprehends microorganisms strictly anaerobic and capable of degrading dichloromethane and was found enriched in both obese and non-obese asthmatic patients [56], whereas *Erysipelotrichaceae cc-115* was found depleted in the gut microbiota of community-dwelling physically active older men [57].

Six different genera were enriched in LG-G12 A (G3) after the end of the experimental period: *Ruminococcus* (LDA > 4), *Anaerotruncus* (LDA > 4), *Bilophila* (LDA > 3), *Desulfovibrio* (LDA > 3), *Brachyspira* (LDA > 2), and *Coprococcus* (LDA > 2). We observed that when LG-G12 was offered to animals that were fed a high-fat diet, there was a remarkable enrichment of SCFA-producing bacteria such as *Ruminococcus*, *Anaerotruncus*, and *Coprococcus* which are commonly found in the microbiota of overweight or obese patients [58,59,60]. Intriguingly, we also detected the enrichment of taxa involved in mucus degradation and hydrogen sulfide production such as *Brachyspira*, *Bilophila*, and *Desulfovibrio*, which may indicate a limited action of LG-G12 against these taxa.

The genera *Enterococcus* (LDA > 5), *Salinispora* (LDA > 3), and *Akkermansia* (LDA > 4) were identified as biomarkers of the ceftriaxone A group (G4), whereas only *Clostridium* (LDA > 4) was significantly enriched in the ceftriaxone + LG-G12 A group (G5). Interestingly, *Akkermansia*, which is a Gram-negative, obligate anaerobe, non-motile, non-spore-forming bacterium, seems to be resilient to the adverse effects of the antimicrobial ceftriaxone. This genus has attracted great interest due to its capability to enhance mucus formation, activate the innate immune system, and promote intestinal homeostasis [61]. As reported by Vesić & Kristich [62], the genus *Enterococcus* is intrinsically resistant to cephalosporins, antibiotics that act on cell wall biosynthesis, which may explain its identification as a biomarker of this group. Additionally, Mishra & Ghosh [63] reports that the *E. faecalis* AG5 strain mitigates HFD-induced obesity through several mechanisms such as activation of adipocyte apoptosis and the improvement of glucose, insulin, and leptin sensitivity.

The genus *Corynebacterium* (LDA > 3) was identified as a biomarker of group G1, while the genera *Blautia* (LDA > 3), *Clostridium* (LDA > 4), and *Akkermansia* (LDA > 5) appear enriched in the control group G6 (AIN-93 intake during the treatment phase). The enrichment of the SCFA-producing bacteria *Akkermansia*, *Blautia*, and *Clostridium* may contribute to restoring intestinal integrity and the development of intestinal homeostasis. During calorie-restricted diet therapy for overweight or obese individuals, insulin resistance improves and is correlated with an increased abundance of *Akkermansia* in the gut [61].

Finally, an enrichment of the *Bifidobacterium* genus (LDA > 4) was observed following LG-G12 with a standard diet in the treatment phase (G7). This result indicates a synergism between LG-G12 and endogenous bifidobacteria, which might be strain-specific. Following probiotic intervention with *Latilactobacillus curvatus* and *Lactiplantibacillus plantarum*, Park et al. [64] observed enrichment of *Bifidobacterium pseudolongum* species in HFD-probiotic mice when compared to the HFD-placebo group. Differentially abundant genera were not identified in the ceftriaxone (G8) and ceftriaxone + LG-G12 groups (G9).

### 3.4. Correlation Analyses

#### 3.4.1. Groups Treated with LG-G12

The genera *Enterococcus* (r = −0.59, *p* = 6.27 × 10^−3^) and *Bifidobacterium* (r = −0.54, *p* = 1.32 × 10^−2^) were negatively correlated with high Lactulose/Mannitol (L/M) ratio (Table 3), in LG-G12 treated groups (G3 and G7) (Figure 4A). Species of the genus *Enterococcus* can interact with mucosal immune cells, thus activating intestinal immune response [64]. Wu et al. [65] report that the probiotic *E. faecium* NCIMB 11181 can ameliorate necrotic enteritis by improving intestinal mucosal barrier function and modulating gut microbiota. *Bifidobacterium*, which is indicative of microbial diversity [66], can protect against obesity and diabetes, as well as improve intestinal integrity and control metabolic endotoxemia, important parameters for the assessment of intestinal balance and health [41,67].

In terms of SCFA production (Figure 4B, Table 3), the genera *Enterococcus* (r = 0.48, *p* = 3.12 × 10^−2^), *Allobaculum* (r = 0.76, *p* = 8.96 × 10^−5^), *Sporosarcina* (r = 0.83, *p* = 4.73 × 10^−6^), *Jeotgalicoccus* (r = 0.87, *p* = 7.67 × 10^−7^), *Staphylococcus* (r = 0.91, *p* = 2.50 × 10^−8^), *Bifidobacterium* (r = 0.60, *p* = 4.79 × 10^−3^), *Blautia* (r = 0.59, *p* = 6.11 × 10^−3^) were positively correlated with the total amount of SCFA, whereas *Prevotella* (r = 0.50, *p* = 2.33 × 10^−2^) was only positively correlated with the production of butyrate.

Kong et al. [41] reported that the consumption of high-fat and high-sucrose diets reduces the abundance of *Prevotella* and, consequently, butyrate levels. In addition, the probiotic administration (*Lactobacillus acidophilus*, *Bifidobacterium longum*, and *Enterococcus faecalis*) can restore the intestinal microbiota, increasing microorganisms such as *Lactobacillus*, *Bifidobacterium*, and *Akkermansia*. Based on our results, we believe that LG-G12 positively modulated SCFA-producing bacteria such as *Allobaculum*, *Bifidobacterium*, and *Prevotella*.

Overall, the bacterial genera that negatively correlated with the L/M ratio were positively correlated with the production of SCFA, indicating a relevant role of such organic acids with the integrity of the intestinal barrier, which is in line with previous studies [41,68,69].

#### 3.4.2. Groups Treated with Ceftriaxone

The correlation analysis performed considering groups G4, and G8 (Appendix A, Table 3) revealed that *Staphylococcus* (r = −0.81, *p* = 7.49 × 10^−4^) was negatively correlated with lactulose, whereas *Clostridium* (r = 0.48, *p* = 3.79 × 10^−2^) was positively correlated with L/M ratio. Zeng et al. [70] reported that lactulose inhibited the effect of *Staphylococcus aureus* due to the production of sialyllactulose, an antimicrobial enzyme capable to cause damage to the *S. aureus* cell membrane, which can be good for intestinal health. The positive correlation between *Clostridium* and a high L/M ratio is not news as some species such as *Clostridium difficile* can secrete toxins with cytotoxic effects on the intestinal epithelium [71].

In terms of SCFA (Appendix A, Table 3), the genera *Allobaculum* (r = 0.67, *p* = 1.59 × 10^−3^) and *Bifidobacterium* (r = 0.54, *p* = 1.72 × 10^−2^), were positively correlated with total SCFA production, whereas only the genus *Stenotrophomonas* (r = −0.48, *p* = 3.76 × 10^−2^) showed a negative correlation. Even following antimicrobial treatment, the genera *Allobaculum* and *Bifidobacterium* were negatively correlated with the L/M ratio and positively correlated with the production of SCFA, which indicate that these genera may act in the maintenance of intestinal integrity and homeostasis as previously described by Kong et al. [41]. Regarding the genus *Stenotrophomonas*, some species belonging to this genus, such as *Stenotrophomonas maltophilia*, are considered pathogenic bacteria [72] and highly resistant to antibiotics [73], which may justify its presence among the ceftriaxone-treated groups.

#### 3.4.3. Groups Treated with Ceftriaxone Followed by LG-G12

Correlation analyses encompassing groups G5 and G9 (Appendix A) revealed that the genus *Desulfovibrio* (r = 0.48, *p* = 3.29 × 10^−2^) was positively correlated with L/M ratio and, consequently, loss of intestinal integrity. *Desulfovibrio* members are frequently elevated in intestinal dysbiosis, causing intestinal permeability and inflammation [74], which is consistent with our findings.

Concerning SCFAs (Appendix A), the genera *Prevotella* (r = 0.49, *p* = 2.98 × 10^−2^) and *Faecalibacterium* (r = 0.50, *p* = 2.44 × 10^−2^) were positively correlated with their total amount (represented here by the sum of acetate, propionate, and butyrate), whereas some genera were positively correlated with only certain compounds, but not with total production, as follows: *Allobaculum* (Acetic: r = 0.88, *p* = 2.95 × 10^−3^; Propionic, r = 0.73, *p* = 2.70 × 10^−4^; Butyric, r = 0.64, *p* = 2.62 × 10^−3^) and *Bifidobacterium* (Acetic, r = 0.81, *p* = 1.71 × 10^−2^; Propionic, r = 0.67, *p* = 1.11 × 10^−3^; Butyric, r = 0.74, *p* = 1.87 × 10^−4^). Acetate and lactate are among the SCFAs produced by *Bifidobacterium* [75], whereas butyrate production in particular is more related to prebiotics, as discussed previously [68]. The identification of *Faecalibacterium* and *Allobaculum*, both genera described as producers of SCFA [41,69]. This association may also improve intestinal health by promoting SCFA-producing genera and, consequently, enhancing the gut microbiota.

### 3.5. Functional Predictions of the Gut Microbiota

During the different interventions evaluated in the current study, we aimed to identify whether microbial metabolic pathways were enriched in the gut microbiota, which could be associated with the effects of a high-fat diet or not. In addition, we focused on identifying metabolic pathways related to SCFA production that can directly impact intestinal health. Taking into account the comparison between groups G7 and G3, 58 functional pathways differed significantly between the groups (Appendix A), and the great majority (approximately 69.0%) were enriched in group G7. Regarding the MetaCyc pathways associated with SCFA production, six metabolic pathways were identified (Figure 5), and four of them were present in group G7 (*Bifidobacterium* shunt, heterolactic fermentation, hexitol fermentation to lactate, formate, ethanol, and acetate). The enrichment of *Bifidobacterium* shunt, which is a classic pathway of carbohydrate metabolisms such as fructose and glucose, can generate compounds serving as an energy source for intestinal epithelial cells [68,75], which explains the greater intestinal integrity noticed in this group. Only the acetyl-Coa fermentation to butanoate II and L-lysine fermentation to acetate and butanoate pathways were enriched in the G3 group.

In the groups that underwent ceftriaxone treatment followed by LG-G12 (groups G5 and G9), only 14 functional pathways differed significantly between both groups (Appendix A) with approximately 64.0% of the features enriched in the G9 group. The enrichment of the super pathway of D-glucarate and D-galactarate degradation in group G5 also demonstrates the use of alternative carbon sources for growth. The use of dicarboxylic acid sugars as growth substrate occurs in many different bacteria but is especially found in Gram-negative bacteria such as members of the *Enterobacteriaceae*, *Moraxellaceae*, and *Pseudomonadaceae* families [76]. Metabolic pathways, *Bifidobacterium* shunt, and heterolactic fermentation, associated with SCFA production were enriched only in group G9 (Appendix A). Similarly, in the G9 group, we also observed the enrichment of the *Bifidobacterium* shunt and heterolactic fermentation pathways, both associated with carbohydrate metabolism [77], which evidence an important role of low-calorie diets in the enrichment of these functions. Interestingly, the super pathway of D-glucarate and D-galactarate degradation and the pathway of purine nucleotide degradation II (aerobic) were enriched in the G5 group. This indicates that this group of microbes is using alternative carbon sources.

Finally, 12 functional pathways differed significantly between groups G4 and G8 (ceftriaxone administration only) (Appendix A), with the vast majority of pathways (approximately 66.0%) being enriched in the G4 group. The main metabolic pathways enriched in the G4 group were associated with menaquinol biosynthesis and de novo nucleotide biosynthesis. The enrichment of menaquinol biosynthesis might be related to the energy processes of bacteria since menaquinones are relevant growth factors for gut microbiota [78,79] Traditionally, the gut microbiota is an important source of purines, which are used in different functions related to the intestinal barrier and innate immunity, being necessary for intestinal protection and health [80]. Since dysbiosis was observed in group G4, it is believed that the enrichment of nucleotide biosynthesis pathways confirms an expansion of specific microbial taxa in this group.

## 4. Conclusions

Consumption of a high-fat diet associated with ceftriaxone was able to reduce microbial diversity. It was observed that LG-G12 had the best effects when it was combined with a low-calorie diet in restoring gut homeostasis. Higher caecal SCFA contributed to increased intestinal integrity. Also, the genera that presented a negative correlation with a high L/M ratio were similar to those that had a positive correlation with total SCFA production. This trend was further confirmed by metagenomic predictions of the gut microbiota. LG-G12 is presented in this study as a novel adjuvant treatment for overweight or obese individuals through gut microbiota modulation and improvement of intestinal health in models undergoing antimicrobial therapy or not.

## Figures and Tables

**Figure 1 foods-12-01092-f001:**
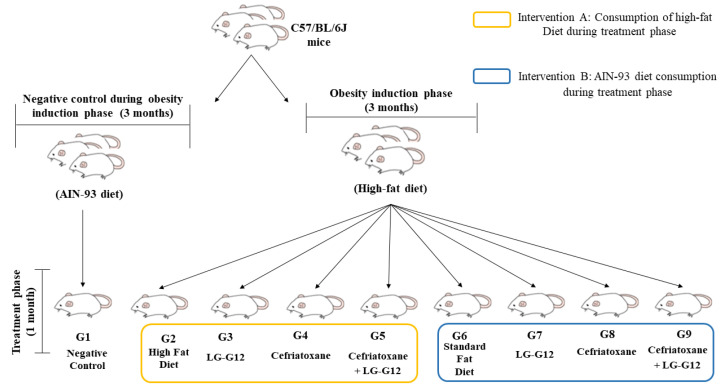
Study Design.

**Figure 2 foods-12-01092-f002:**
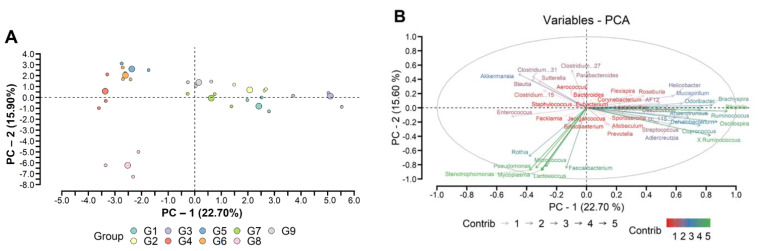
Principal Component Analysis (PCA) (**A**) and factorial analysis (**B**) are based on the most abundant OTUs (relative abundance >0.1%) at the genus level across the groups. G1: negative control group, G2: high fat diet (HFD), G3: LG-G12 HFD, G4: cefriatoxane HFD, G5: cefriatoxane + LG-G12 HFD, G6: standard fat diet (SFD), G7: LG-G12 SFD, G8: cefriatoxane SFD, G9: cefriatoxane + LG-G12 SFD. The red-to-green gradient indicates a low to high magnitude of the contribution of a given genus for a specific factor.

**Figure 3 foods-12-01092-f003:**
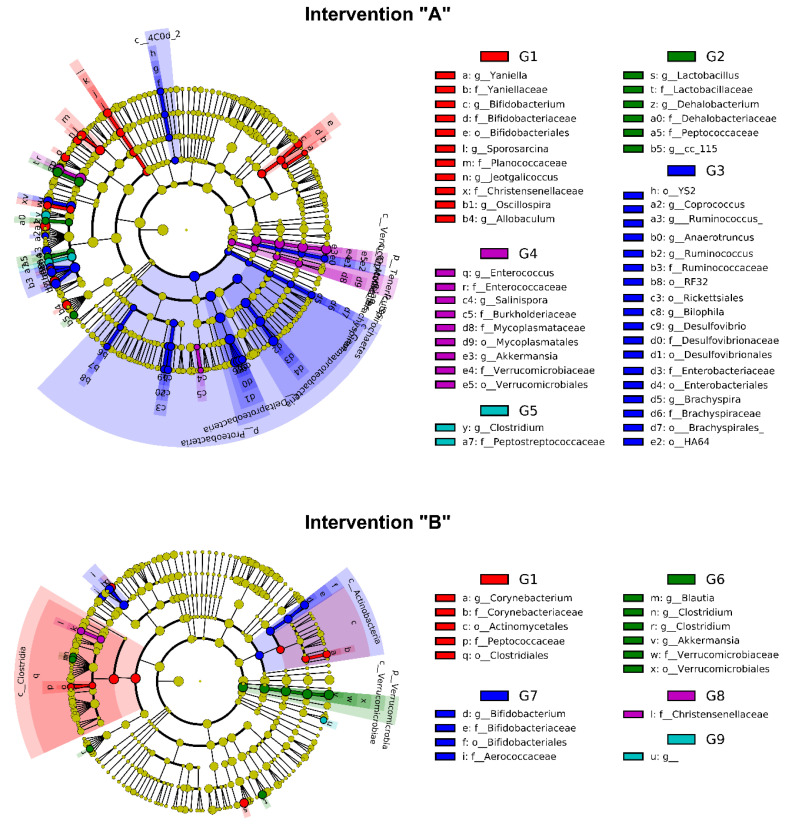
Linear discriminant analysis effect size (LEfSe) analysis. G1: negative control group, G2: high fat diet (HFD), G3: LG-G12 HFD, G4: cefriatoxane HFD, G5: cefriatoxane + LG-G12 HFD, G6: standard fat diet (SFD), G7: LG-G12 SFD, G8: cefriatoxane SFD, G9: cefriatoxane + LG-G12 SFD. Only taxonomic groups showing linear discriminant analysis (LDA) scores > 2.0 with false discovery rate (FDR) *p* < 0.05 are shown. Letters: p, phylum; c, class; o, order; f, family; g, genus. This analysis was carried out to identify significant differences in abundant taxa (fecal biomarkers) after interventions “A” and “B”.

**Figure 4 foods-12-01092-f004:**
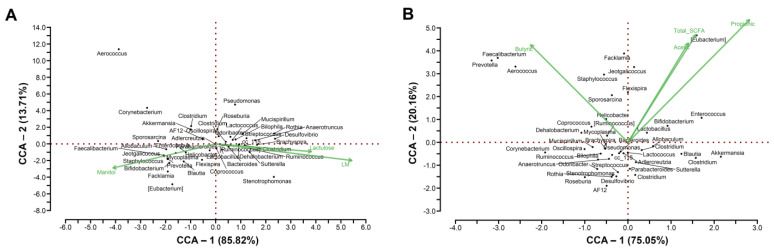
Canonical correspondence analysis (CCA) among groups treated with LG-G12. CCA was performed on the most abundant OTUs (relative abundance > 0.1%) at the genus level for the groups G3 and G7, with (**A**) the intestinal permeability parameters (lactulose, mannitol, and lactulose/mannitol ratio) and (**B**) the short-chain fatty acids acetate, propionate, butyrate, and total SCFA. Green lines indicate the direction and magnitude of measurable variables associated with community structures.

**Figure 5 foods-12-01092-f005:**
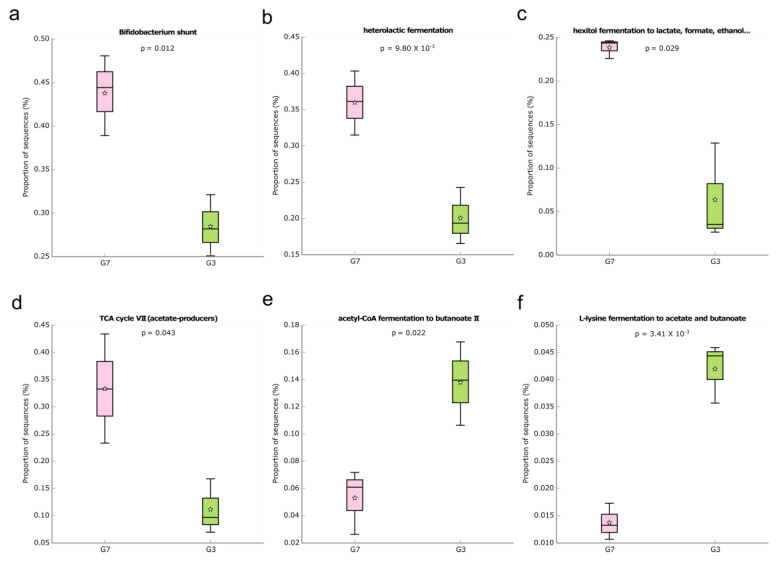
Box plot showing the distribution in the proportion of the predicted MetaCyc pathways related to SCFA (**a**–**f**) production between the groups G3 and G7. Boxes indicate the IQR (75th to 25th of the data). The median value is shown as a line within the box and the mean value as a star. Whiskers extend to the most extreme value within 1.5*IQR. Where: (**a**): *Bifidobacterium* shunt; (**b**): Heterolactic fermentation; (**c**): Hexitol fermentation to lactate, formate, ethanol; (**d**): TCA cycle VII; (**e**): Acetyl-CoA fermentation to butanoate II; (**f**): L-lysine fermentation to acetate and butanoate.

**Table 1 foods-12-01092-t001:** Alpha-diversity indexes of fecal samples were obtained across the different groups after their respective intervention in phase 2 (treatment phase).

Group	Shannon	Chao1	Faith_pd	Observed Features
G1	6.4 ± 0.1 ^A^	394 ± 7.0 ^A^	26.4 ± 2.8	393.3 ± 6.3 ^A^
G2	6.1 ± 0.3 ^A^	363.3 ± 11.2 ^A^	26.5 ± 1.9	363.3 ± 11.2 ^A^
G3	6.2 ± 0.1 ^A^	352.7 ± 16.9 ^A^	24.8 ± 1.7	351.0 ± 18.0 ^A^
G4	2.1 ± 0.8 ^B^	124.7 ± 27.6 ^B^	11.7 ± 1.8	124.0 ± 27.8 ^B^
G5	4.7 ± 0.4 ^AB^	192 ± 50.1 ^AB^	22.6 ± 3.2	228.5 ± 59.5 ^AB^
G6	3.7 ± 0.5 ^AB^	217 ± 20.2 ^AB^	27.6 ± 1.6	233.0 ± 19.0 ^AB^
G7	5.9 ± 0.2 ^A^	338.3 ± 15.9 ^A^	25.3 ± 3.7	337.6 ± 15.7 ^AB^
G8	3.5 ± 1.5 ^AB^	250.5 ± 79.9 ^AB^	22.3 ± 6.5	249.5 ± 138.5 ^AB^
G9	5.2 ± 0.6 ^AB^	270 ± 69.64 ^AB^	24.8 ± 5.7	270.0 ± 69.6 ^AB^

G1: negative control group, G2: high fat diet (HFD), G3: LG-G12 HFD, G4: cefriatoxane HFD, G5: cefriatoxane + LG-G12 HFD, G6: standard fat diet (SFD), G7: LG-G12 SFD, G8: cefriatoxane SFD, G9: cefriatoxane + LG-G12 SFD. Different superscript letters (A and B) indicate significant differences among the groups (*p* < 0.05). The absence of a letter means that there was no statistical difference between the groups (*p* > 0.05). ANOVA followed by Tukey’s test was used in samples with parametric distribution and Kruskal Wallis analysis followed by Dunns’ test in non-parametric samples.

**Table 2 foods-12-01092-t002:** *Firmicutes* to *Bacteroidetes* ratio (F/B), and relative abundance of Gram-negative/positive genera (>0.1%), Mycoplasma, and low abundant genera (LAG) across the different groups.

Group	F/B Ratio	Gram-Negative	Gram-Positive	G+/G− Ratio	*Mycoplasma*	LAG
G1	3.40 ^B^	62.03% ^A^	37.80% ^A^	0.61	0.01%	0.16% ^A^
G2	1.72 ^B^	74.51% ^A^	25.37% ^A^	0.34	0.00%	0.12% ^A^
G3	2.41 ^B^	77.21% ^A^	22.35% ^A^	0.29	0.00%	0.44% ^A^
G4	16.81 ^A^	23.50% ^AB^	76.23% ^AB^	3.24	0.07%	0.21% ^AB^
G5	0.96 ^B^	76.56% ^A^	23.25% ^A^	0.30	0.00%	0.19% ^A^
G6	1.46 ^B^	81.64% ^AC^	18.29% ^AC^	0.22	0.01%	0.07% ^AC^
G7	2.06 ^B^	49.92% ^A^	49.96% ^A^	1.00	0.02%	0.11% ^A^
G8	1.18 ^B^	42.74% ^A^	39.36% ^A^	0.92	14.95%	2.94% ^A^
G9	2.07 ^B^	65.15% ^A^	34.63% ^A^	0.53	0.06%	0.16% ^A^

G1: negative control group, G2: high fat diet (HFD), G3: cefriatoxane HFD, G4: LG-G12 HFD, G5: cefriatoxane + LG-G12 HFD, G6: standard fat diet (SFD), G7: cefriatoxane SFD, G8: LG-G12 SFD, G9: cefriatoxane + LG-G12 SFD. Different superscript letters (A, B, and C) indicate significant differences among the groups (*p* < 0.05). ANOVA followed by Tukey’s test was used in samples with parametric distribution and Kruskal Wallis analysis followed by Dunns’ test in non-parametric samples.

**Table 3 foods-12-01092-t003:** Short-chain fatty acid (SCFA) concentration (µmol/g feces) in caecal samples and lactulose/mannitol ratio (L/M) obtained across the different groups after their respective intervention in phase 2 (treatment phase).

Group	Acetate	Propionate	Butyrate	Total SCFA	L/M
G1	571.80 ± 37.40 ^A^	150.70 ± 17.40 ^A^	31.35 ± 8.90 ^A^	753.80 ± 61.00 ^A^	1.41 ± 0.22 ^C^
G2	0.19 ± 0.02 ^D^	0.04 ± 0.01 ^B^	0.05 ± 0.04 ^B^	0.28 ± 0.04 ^C^	55.10 ± 2.25 ^A^
G3	0.16 ± 0.03 ^D^	0.04 ± 0.01 ^B^	0.01 ± 0.01 ^B^	0.19 ± 0.03 ^C^	49.30 ± 3.20 ^A^
G4	0.06 ± 0.02 ^D^	0.03 ± 0.01 ^B^	-	0.10 ± 0.03 ^C^	59.60 ± 3.02 ^A^
G5	0.20 ± 0.08 ^D^	0.07 ± 0.03 ^B^	0.02 ± 0.01 ^B^	0.27 ± 0.07 ^C^	27.50 ± 6.70 ^B^
G6	382.80 ± 44.30 ^B^	142.20 ± 28.20 ^A^	7.20 ± 3.60 ^B^	530.70 ± 68.24 ^AB^	2.10 ± 0.80 ^C^
G7	374.70 ± 61.90 ^B^	140.70 ± 17.70 ^A^	40.10 ± 4.00 ^A^	535.40 ± 80.13 ^AB^	0.98 ± 0.17 ^C^
G8	165.30 ± 23.40 ^C^	185.70 ± 17.30 ^A^	-	351.00 ± 31.00 ^B^	0.58 ± 0.21 ^C^
G9	396.00 ± 45.20 ^B^	211.20 ± 27.90 ^A^	36.70 ± 6.90 ^A^	644.00 ± 68.70 ^A^	2.00 ± 0.23 ^C^

G1: negative control group, G2: high fat diet (HFD), G3: cefriatoxane HFD, G4: LG-G12 HFD, G5: cefriatoxane + LG-G12 HFD, G6: standard fat diet (SFD), G7: cefriatoxane SFD, G8: LG-G12 SFD, G9: cefriatoxane + LG-G12 SFD. Total SCFA refers to the sum of acetate, propionate, and butyrate. Different superscript letters (A, B, C, and D) indicate significant differences among the groups (*p* < 0.05). ANOVA followed by Tukey’s test was used in samples with parametric distribution and Kruskal Wallis analysis followed by Dunns’ test in non-parametric samples.

## Data Availability

The data presented in this study are available on request from the corresponding author. The data are not publicly available due to privacy restrictions.

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
