# Peer review of "Lactobacillus gasseri* LG-G12 Restores Gut Microbiota and Intestinal Health in Obesity Mice on Ceftriaxone Therapy"

_foods, 2023, doi:10.3390/foods12051092_

Round 1
Reviewer 1 Report
The current article describes the potential use of Lactobacillus gasseri LG-G12 as a probiotic to manage obesity by restoring gut microbiota and intestinal health.
Methods:
By pooling samples, does that mean that fecal samples from 7 animals were pooled into one sample? Could authors provide more details about this? Some groups have been 8 animals, rather than 7 animals.
Were there any differences in food intake between different cages of animals?
For the functional analysis part, were there differences between G2 and G6? Any comparison of these groups to G3 and G7 as changes in the diet might already trigger changes?
General comments:
Please ensure that scientific names are written in italics.
Positive control: should be HFD model group?
Page 3, Line 105: change “intervantions” to “interventions”
Heading 3.0 should be "Results and discussion".
The article presents interesting findings but it would be great to also explain on differences in gut microbiota (e.g. the dynamics of microbiota) between G1, G2 and G6 - especially on the differences through functional analyses. This would certainly help to emphasize the potential of probiotics in the management of obesity.
Author Response
January 26th, 2023.
Dear Fannie Fang
Editor
Thank you for considering the manuscript entitled “Lactobacillus gasseri LG-G12 restores gut microbiota and intestinal health in obesity mice on ceftriaxone therapy” for publication.
We thank for the comments and considerations, and we inform that all requested changes were made. In the new version of the manuscript the changes have been marked in red.
We emphasize that the intention of this work is to carry out a broad comparison between the groups, since the data from specific interventions have already been published previously (Dias et al. 2021 and Louzano et al. 2022).
On behalf of all authors, I hope that the changes can match the requests in the best way possible. Find below the answers to the reviewers’ comments.
Best regards,
Mariana de Moura e Dias
Responding to comments
# Reviewer 1:
Methods:
- By pooling samples, does that mean that fecal samples from 7 animals were pooled into one sample? Could authors provide more details about this? Some groups have been 8 animals, rather than 7 animals.
Answer: Yes, that’s right. The fecal samples from all animals belonging to the same group were pooled into one sample, metagenomic DNA extracted and sequenced (triplicate). Unfortunately, the rationale for that was the limited financial resources conduct amplicon sequencing for 216 samples (nine groups, eight animals/group, triplicate). The reason why some groups had 7 animals is because, in total, 6 animals died of natural causes during the experimental and weren’t replaced.
- Were there any differences in food intake between different cages of animals?
Answer: There were no differences for food intake between groups. In addition, the animals were submitted to the pair feeding scheme.
- For the functional analysis part, were there differences between G2 and G6? Any comparison of these groups to G3 and G7 as changes in the diet might already trigger changes?
Answer: We would like to thank #reviewer1 for this inquiry. For the functional analyzes we focused only on the groups that underwent interventions with LG-G12 (G3 vs G7), LG-G12 associated with ceftriaxone (G5 vs G9), and only ceftriaxone (G4 vs G8) due to the focus on the production of short chain fatty acids (SCFAs) and their beneficial effect on intestinal health. Thus, the other groups were not mentioned due to the lack of enrichment and consequent absence of statistical significance in these pathways. Therefore, we hypothesized that the enrichment in metabolic pathways associated with the production of SCFAs was mostly due to the administration of the probiotic candidate LG-G12 and its modulatory effect on the intestinal microbiota.
General comments:
- Please ensure that scientific names are written in italics.
Answer: Change made as requested. All scientific names are in italics.
- Positive control: should be HFD model group?
Answer: This change was made. “Positive control A” group was changed to “high fat diet” and “positive control B” was changed to “standard fat diet”
- Page 3, Line 105: change “intervantions” to “interventions”
Answer: Change made as requested.
- Heading 3.0 should be "Results and discussion".
Answer: Change made as requested.
The article presents interesting findings but it would be great to also explain on differences in gut microbiota (e.g. the dynamics of microbiota) between G1, G2 and G6 - especially on the differences through functional analyses. This would certainly help to emphasize the potential of probiotics in the management of obesity.
Answer: Thanks again to the interesting comment. The dynamics based on amplicon sequencing was alrealdy published by Dias et al. 2021 and Louzano et al. 2022, reason why hasn’t been mentioned in this study.
With regards to the functional analysis, currently we are working on acquiring financial resources to conduct shotgun sequencing of these samples. We aim not only to delve deeper into the metabolic pathways of the microbial communities, but also work on their metagenome-assembled genomes. This will certainly help to have a better overview of the microbial communities and probiotic/diet impact in the management of obesity.
References:
- Dias, M. M.; Louzano, S. A. R.; Conceição, L. L.; Conceição, R. F.; Mendes, T. A. O.; Pereira, S. S.; Oliveira, L. L. Peluzio, M. C. G. Antibiotic Followed by a Potential Probiotic Increases Brown Adipose Tissue, Reduces Biometric Measurements, and Changes Intestinal Microbiota Phyla in Obesity. Probiotics Antimicrob Proteins 2021, 13, 1621-1631.
- Louzano, S. A. R.; Dias, M. M.; Conceição, L. L.; Mendes, T. A. O.; Peluzio, M. C. G. Ceftriaxone causes dysbiosis and changes intestinal structure in adjuvant obesity trea-tment. Pharmacol Rep 2022, 74, 111-123.

Reviewer 2 Report
In the present manuscript the authors describe the effects of the probiotic strain L. gasseri LG-G12, ceftriaxone, and diet on the gut microbiota. The authors report significant differences in gut microbiota composition and SCFA profile induced by the probiotic strain and conclude that this this strain was able to restore the gut homeostasis.
Generally, the scope of the study is interesting, and the manuscript is clearly written. However, several issues needed to be addressed. Specific comments are listed below.
Methods
1. The authors do not explain why they chose L. gasseri LG-G12for their study. Are there any other studies supporting its potential positive role in gut microbiota?
2. Change the names of positive control groups A and B to make it easier for the reader to understand the content of the paper. I suggest using HFD (high fat diet) instead of positive control A and SFD (standard fat diet) instead of positive control B.
3. Technical replicates for the sequencing would have given more reliable results instead of the pool of samples. What is the rationale for such approach?
Results
4. The dominant species identified at different region of colons should be displayed as a figure or table.
5. Lines 228-229: The authors stated that there were no differences regarding alpha diversity indices between the interventions “A” and “B. However, according to Table 1, there were some differences among groups.
6. Section 3.2 – Even though the title refers to beta diversity, there is no result regarding this evaluation.
7. Lines 285-285: “As expected, the G4 group showed the least proportion of Gram-negative among all groups (23.50%) and the greatest G+/G- ratio (3.24)…” Was this expected for the G8 as well? This should be addressed by the authors.
8. Tables 2 and 3: the meanings of the G3 x G4 and G7 x G8 groups are interchanged.
9. Table 2: I suggest using the same layout adopted for table 1 (names of groups displayed in colum).
10.I consider more appropriate to indicate in the legend of each table which analysis was performed, as well as the p-values that were obtained.
11.Lines 358-361: “When the biomarkers associated with the intervention B are analyzed, fewer genera are observed when compared to the intervention A, which reinforces the strong effect of the diet on the reestablishment of the dysbiotic gut microbiota in the groups that underwent a high-fat diet.” Considering that loss of diversity may be related to dysbiosis, this sentence needs to be revised.
12.Considering that an elevated lactulose to mannitol ratio is an indicator of intestinal barrier dysfunction, please explain why some groups from G6-G9 showed lower L/M ratio thanG1 group and than G2-G5 groups (Table 3).
13.Most of the discussions are either not related to the current study or deviated from the results obtained. In addition, it seems the discussions are written in the form literature review rather than discussing the results obtained from the current study. A thorough revision on this section is required.
Conclusion
14.The conclusion should be rewritten as it does not conclude based on the results obtained. For example, the author informed that “High-fat diet intake remarkably impaired bacterial diversity indices,”. However, the diversity found in G2 and G3 were superior than those found on their correspondent groups from normal diet (G6 and G7, respectively).
Minor suggestions:
Lines 116 and 119: 109 instead of 109
Line 118: 500 mg of ceftriaxone per kg of body or 500 mg of ceftriaxone/kg of body instead of 500 mg of ceftriaxone kg of body
Line 210: Results and discussion instead of Results
The letters used to indicate statistical differences should be standardized so that the letter "a" is used for the highest value.
All microorganisms’ genera and species need to be italicized.
Finally, I suggest bringing the manuscript under a specialized language revision since it still presents grammatical and spelling mistakes. I include some of them below:
Line 23: restore instead of restores
Line 48: modulate instead of modulates
Line 51: synbiotic instead of symbiotic
Line 80: affecting instead of affect
Author Response
January 26th, 2023.
Dear Fannie Fang
Editor
Thank you for considering the manuscript entitled “Lactobacillus gasseri LG-G12 restores gut microbiota and intestinal health in obesity mice on ceftriaxone therapy” for publication.
We thank for the comments and considerations, and we inform that all requested changes were made. In the new version of the manuscript the changes have been marked in red.
We emphasize that the intention of this work is to carry out a broad comparison between the groups, since the data from specific interventions have already been published previously (Dias et al. 2021 and Louzano et al. 2022).
On behalf of all authors, I hope that the changes can match the requests in the best way possible. Find below the answers to the reviewers’ comments.
Best regards,
Mariana de Moura e Dias
Responding to comments
# Reviewer 2:
Methods
- The authors do not explain why they chose L. gasseri LG-G12 for their study. Are there any other studies supporting its potential positive role in gut microbiota?
Answer: Since the initial objective of the project was to study an adjuvant treatment for obesity, Lactobacillus gasseri was chosen for its effects on weight loss and body fat reduction, as briefly described in lines 51 and 52. More information was not provided in this manuscript since the justification for the choice and the potential of LG-G12 have already been published by Dias et al. 2021. Also, as described by Dias et. al. 2021, there are other studies using different strains of Lactobacillus gasseri that obtained positive results for the treatment of obesity.
In brief, different strains of L. gasseri show positive results in controlling excess weight and body adiposity. In addition, its origin is human and it has the ability to survive to intestinal transit. Finally, different strains of L. gasseri decrease systemic inflammation and intestinal permeability, which is a mechanism used by probiotics to treat obesity.
- Change the names of positive control groups A and B to make it easier for the reader to understand the content of the paper. I suggest using HFD (high fat diet) instead of positive control A and SFD (standard fat diet) instead of positive control B.
Answer: Change made as requested.
- Technical replicates for the sequencing would have given more reliable results instead of the pool of samples. What is the rationale for such approach?
Answer: Unfortunately, the rationale for that was the limited financial resources to conduct amplicon sequencing for 216 samples (nine groups, eight animals/group, triplicate).
Results
- The dominant species identified at different region of colons should be displayed as a figure or table.
Answer: We apologize, but it is not possible to present species from different regions of the colon, as we only work with feces.
- Lines 228-229: The authors stated that there were no differences regarding alpha diversity indices between the interventions “A” and “B. However, according to Table 1, there were some differences among groups.
Answer: Yes, we agree with #reviewer2 that some differences among groups were observed, specially between G4 vs G1, G2 and G3 (Intervention A), as well as for G4 vs G7 (Intervention B). However, this statement refers to shifts in such indices within the same groups (e.g., G4 and G8) before under administration of a high-fat diet (Intervention A) and after returning to the diet AIN-93 (Intervention B). Therefore, we observed an absence of statistical difference (p>0.05) for all pair-wise comparisons within groups in the different intervention.
- Section 3.2 – Even though the title refers to beta diversity, there is no result regarding this evaluation.
Answer: We apologize for this mistake. Beta diversity analysis was included and discussed as well (lines: 248 to 260).
- Lines 285-285: “As expected, the G4 group showed the least proportion of Gram-negative among all groups (23.50%) and the greatest G+/G- ratio (3.24)…” Was this expected for the G8 as well? This should be addressed by the authors.
Answer: This result is not expected for the G8, given that the diet consumed is different in G4 and G8 group. As stated in the manuscript, the diet also modulates the intestinal microbiota, and its effect must be considered.
- Tables 2 and 3: the meanings of the G3 x G4 and G7 x G8 groups are interchanged.
Answer: We are sorry, but we do not understand what change should be made.
- Table 2: I suggest using the same layout adopted for table 1 (names of groups displayed in colum).
Answer: Change made as requested.
- I consider more appropriate to indicate in the legend of each table which analysis was performed, as well as the p-values that were obtained.
Answer: ANOVA followed by Tukey's test was performed for samples with parametric distribution and Kruskal Wallis analysis followed by Dunns' test for samples with non-parametric distribution. This information was added to the manuscript. P values were not added since there are nine groups, and comparisons were made between pairs.
- Lines 358-361: “When the biomarkers associated with the intervention B are analyzed, fewer genera are observed when compared to the intervention A, which reinforces the strong effect of the diet on the reestablishment of the dysbiotic gut microbiota in the groups that underwent a high-fat diet.” Considering that loss of diversity may be related to dysbiosis, this sentence needs to be revised.
Answer: We apologize. This sentence was taken from the manuscript.
- Considering that an elevated lactulose to mannitol ratio is an indicator of intestinal barrier dysfunction, please explain why some groups from G6-G9 showed lower L/M ratio than G1 group and than G2-G5 groups (Table 3).
Answer: For G1, G6 and G9 there is no statistical difference, that is, G6 and G9 are not smaller than G1, but equal. For G2 and G5, the highest values are expected given that they consumed a high-fat diet throughout the experiment. Therefore, it is expected that in these groups (G2-G5) there is indeed a greater dysfunction of the intestinal barrier.
- Most of the discussions are either not related to the current study or deviated from the results obtained. In addition, it seems the discussions are written in the form literature review rather than discussing the results obtained from the current study. A thorough revision on this section is required.
Answer: We thank to the reviewer for the comment. We declare that we agree and try to resolve this comment, but there is still a gap in the literature. We performed new searches in the literature with the terms "obesity", "gut microbiota", "ceftriaxone" and "Lactobacillus gasseri LG-G12", but the new studies found were those of our own research group.
Conclusion
- The conclusion should be rewritten as it does not conclude based on the results obtained. For example, the author informed that “High-fat diet intake remarkably impaired bacterial diversity indices,”. However, the diversity found in G2 and G3 were superior than those found on their correspondent groups from normal diet (G6 and G7, respectively).
Answer: The conclusion has been rewritten as requested. We clarify that we consider that diversity was affected by the high-fat diet and ceftriaxone since G4 was the only group that was distinguished from G1, which was the control. For G2, G3, G6 and G7 there were no significant differences between groups.
Minor suggestions:
- Lines 116 and 119: 109 instead of 109
Answer: Change made as requested.
- Line 118: 500 mg of ceftriaxone per kg of body or 500 mg of ceftriaxone/kg of body instead of 500 mg of ceftriaxone kg of body
Answer: Change made as requested.
- Line 210: Results and discussion instead of Results
Answer: Change made as requested.
- The letters used to indicate statistical differences should be standardized so that the letter "a" is used for the highest value.
Answer: Change made as requested.
- All microorganisms’ genera and species need to be italicized.
Answer: Change made as requested. All scientific names are in italics.
- Finally, I suggest bringing the manuscript under a specialized language revision since it still presents grammatical and spelling mistakes. I include some of them below:
Answer: We appreciate the suggestion and declare that the entire text has been revised in English.
Line 23: restore instead of restores
Answer: Change made as requested.
Line 48: modulate instead of modulates
Answer: Change made as requested.
Line 51: synbiotic instead of symbiotic
Answer: Change made as requested.
Line 80: affecting instead of affect
Answer: Change made as requested.
References:
- Dias, M. M.; Louzano, S. A. R.; Conceição, L. L.; Conceição, R. F.; Mendes, T. A. O.; Pereira, S. S.; Oliveira, L. L. Peluzio, M. C. G. Antibiotic Followed by a Potential Probiotic Increases Brown Adipose Tissue, Reduces Biometric Measurements, and Changes Intestinal Microbiota Phyla in Obesity. Probiotics Antimicrob Proteins 2021, 13, 1621-1631.
- Louzano, S. A. R.; Dias, M. M.; Conceição, L. L.; Mendes, T. A. O.; Peluzio, M. C. G. Ceftriaxone causes dysbiosis and changes intestinal structure in adjuvant obesity trea-tment. Pharmacol Rep 2022, 74, 111-123.

Reviewer 3 Report
Authors have carried out a very interesting study about the effect of probiotics (LG-G12 ), antimicrobials (ceftriaxone) and diet (AIN-93 standard diet) on the gut microbiota of obese mice.
Some minor corrections must be done:
*revise English language: (2.2 Methods, Fig 1.: not "intervention" nor "intervation"; Table 2: ..bacteriodetes; )
*Table 3: uniformice significant decimals; Table 2: double dot
*Line 435: provoking is not the usual word; line 434: revise the need of italics all over the text
*revise References: write in italics (name of journals 22, 37 etc., name of microorganisms 70, 77; etc.); ref 36 not &; ref 30 words in the title in lower case )
Author Response
January 26th, 2023.
Dear Fannie Fang
Editor
Thank you for considering the manuscript entitled “Lactobacillus gasseri LG-G12 restores gut microbiota and intestinal health in obesity mice on ceftriaxone therapy” for publication.
We thank for the comments and considerations, and we inform that all requested changes were made. In the new version of the manuscript the changes have been marked in red.
On behalf of all authors, I hope that the changes can match the requests in the best way possible. Find below the answers to the reviewers’ comments.
Best regards,
Mariana de Moura e Dias
Responding to comments
# Reviewer 3:
- Revise English language: (2.2 Methods, Fig 1.: not "intervention" nor "intervation"; Table 2: ..bacteriodetes;)
Answer: We declare that the entire text has been revised in English.
- Table 3: uniformice significant decimals; Table 2: double dot
Answer: Change made as requested. Table 3 is with double dot too.
- Line 435: provoking is not the usual word;
Answer: We appreciate the suggestion and declare that the entire text has been revised in English.
- Line 434: revise the need of italics all over the text
Answer: Change made as requested. All scientific names are in italics.
- Revise References: write in italics (name of journals 22, 37 etc., name of microorganisms 70, 77; etc.); ref 36 not &; ref 30 words in the title in lower case)
Answer: We appreciate the suggestion and declare that references were checked.

Round 2
Reviewer 2 Report
The authors answered to all questions raised. But I have just a few more remarks about the tables and figures:
In all tables and figures (that apply),
- cefriatoxane A and cefriatoxane B should be changed to cefriatoxane high fat diet or HFD and cefriatoxane standard fat diet or SFD.
- LG-G12 A and LG-G12 B should be changed to LG-G12 high fat diet or HFD and LG-G12 standard fat diet or SFD.
This is necessary to match to Figure 1.
- In Tables 2 and 3, the captions are:
"G1: negative control group, G2: high fat diet, G3: cefriatoxane A, G4: LG-G12 A, G5: 446 cefriatoxane + LG-G12 A, G6: standard fat diet, G7: cefriatoxane B, G8: LG-G12 B, G9: 447 cefriatoxane + LG-G12 A."
But G3 should be LG-G12 HFD, G4 should be cefriatoxane HFD, G7 should be LG-G12 SFD and G8 should be cefriatoxane SFD
Author Response
January 31th, 2023.
Dear Fannie Fang
Editor
Thank you for considering the manuscript entitled “Lactobacillus gasseri LG-G12 restores gut microbiota and intestinal health in obesity mice on ceftriaxone therapy” for publication.
We thank for the comments and considerations, and we inform that all requested changes were made. In the new version of the manuscript the changes have been marked in red.
On behalf of all authors, I hope that the changes can match the requests in the best way possible. Find below the answers to the reviewers’ comments.
Best regards,
Mariana de Moura e Dias
Responding to comments
# Reviewer 2:
The authors answered to all questions raised. But I have just a few more remarks about the tables and figures:
In all tables and figures (that apply),
- cefriatoxane A and cefriatoxane B should be changed to cefriatoxane high fat diet or HFD and cefriatoxane standard fat diet or SFD.
- LG-G12 A and LG-G12 B should be changed to LG-G12 high fat diet or HFD and LG-G12 standard fat diet or SFD.
This is necessary to match to Figure 1.
Answer: Thanks to the reviewers for the comment. The changes were made in figures 2 and 3.
- In Tables 2 and 3, the captions are:
"G1: negative control group, G2: high fat diet, G3: cefriatoxane A, G4: LG-G12 A, G5: 446 cefriatoxane + LG-G12 A, G6: standard fat diet, G7: cefriatoxane B, G8: LG-G12 B, G9: 447 cefriatoxane + LG-G12 A."
But G3 should be LG-G12 HFD, G4 should be cefriatoxane HFD, G7 should be LG-G12 SFD and G8 should be cefriatoxane SFD
Answer: Thanks to the reviewers for the comment. The changes were made in table 1, 2 and 3.
